# Disentangling top-down drivers of mortality underlying diel population dynamics of *Prochlorococcus* in the North Pacific Subtropical Gyre

Stephen J. Beckett [1,2,24] ✉, David Demory [1,3,24] ✉, Ashley R. Coenen[4], John R. Casey[5,6,7,8], Mathilde Dugenne[5,6,9], Christopher L. Follett [7,10], Paige Connell[11,12], Michael C. G. Carlson [13,14], Sarah K. Hu[11,15,16], Samuel T. Wilson[5,6,17], Daniel Muratore [1,18], Rogelio A. Rodriguez-Gonzalez[1], Shengyun Peng[1,19], Kevin W. Becker [15,20], Daniel R. Mende[5,6,21], E. Virginia Armbrust [22], David A. Caron[11], Debbie Lindell [13], Angelicque E. White [5,6], François Ribalet [22] & Joshua S. Weitz [1,2,4,23] ✉

Photosynthesis fuels primary production at the base of marine food webs. Yet, in many surface ocean ecosystems, diel-driven primary production is tightly coupled to daily loss. This tight coupling raises the question: which top-down drivers predominate in maintaining persistently stable picocyanobacterial populations over longer time scales? Motivated by high-frequency surface water measurements taken in the North Pacific Subtropical Gyre (NPSG), we developed multitrophic models to investigate bottom-up and top-down mechanisms underlying the balanced control of *Prochlorococcus* populations. We find that incorporating photosynthetic growth with viral- and predator-induced mortality is sufficient to recapitulate daily oscillations of *Prochlorococcus* abundances with baseline community abundances. In doing so, we infer that grazers in this environment function as the predominant top-down factor despite high standing viral particle densities. The model-data fits also reveal the ecological relevance of light-dependent viral traits and non-canonical factors to cellular loss. Finally, we leverage sensitivity analyses to demonstrate how variation in life history traits across distinct oceanic contexts, including variation in viral adsorption and grazer clearance rates, can transform the quantitative and even qualitative importance of top-down controls in shaping *Prochlorococcus* population dynamics.

Highly resolved surface ocean observations reveal repeatable daily changes in the abundance of ubiquitous picocyanobacteria at the base of the marine microbial food web, including *Prochlorococcus* and *Synechococcus*[1,2]. Typically, picocyanobacteria decrease in abundance during the day and then increase overnight. Oscillatory phytoplankton population dynamics are influenced by nutrient and light availability[2–10] and by density- and size-dependent feedback processes with other community components[11–17]. As a result, these interactions

lead to diel oscillations in related ecological processes, including grazing rates, viral infection rates, and viral activity[18–23]. The presence of diel oscillations often make it challenging to infer process from pattern, e.g., reduced population growth and/or increased mortality can have the same net effect on abundances[2,10–12].

Across oceanic basins, grazers and viruses are hypothesized to be the dominant drivers of phytoplankton loss[24,25]. However, estimating the relative contribution of viral-induced and grazing-induced mortality at a particular site remains challenging in the absence of additional ecosystem-specific process information[22,26–29]. We focus our analysis on a near-surface Lagrangian parcel of water in the North Pacific Subtropical Gyre (NPSG), sampled at high temporal resolution at 15-m depth over 10 days in summer 2015 by the SCOPE HOE-Legacy 2A cruise (see "Methods"). The oligotrophic NPSG is numerically dominated by the unicellular cyanobacterium *Prochlorococcus*, the most abundant photosynthetic organism in the global oceans[30,31]. Prior work using the cellular iPolony method estimates that cyanophage, despite being highly abundant, contribute to <5% of total *Prochlorococcus* cellular losses per day[22]. In parallel, analysis of food requirements to maintain heterotrophic nanoflagellate abundances suggest that grazing could account for the majority of daily *Prochlorococcus* cell losses[23]. However, this quota method cannot rule out potentially significantly lower rates of grazing, especially if *Prochlorococcus* represent only a part of the diet of heterotrophic nanoflagellates. Grazing is expected to drive the flow of matter through marine food webs and out of the surface ocean ecosystem via export and subsequent sinking of fecal pellets[32]. In contrast, viral infection and lysis are expected to shunt matter back into the microbial loop[33–35], though significant lysis (e.g., during blooms) may lead to sticky aggregate production and increased export out of the surface ocean[36,37]. Hence, disentangling the relative rates of viral-induced lysis and grazing can help inform estimates of the link between primary production and export.

Here, we use an ecological modeling and statistical fitting framework, combined with field observations, as a means to understand how observed *Prochlorococcus* dynamics are shaped by a combination of bottom-up and top-down forces in the NPSG. The multi-trophic models combine principles of virus-microbe interactions and grazing[29,38–41] with light-driven forcing of cellular physiology[2,11]. Using Bayesian Markov chain Monte Carlo fitting methods, we compare in silico model dynamics with measured in situ ecological rhythms. We then use model-data fits across a range of ecological scenarios as a means to robustly estimate the contribution of viral-induced lysis and grazing to total *Prochlorococcus* mortality. As we show, model-data integration suggests the tight coupling of *Prochlorococcus* growth and loss over diel cycles in the NPSG is due primarily to the impact of grazing—and not viral-induced mortality. In doing so, we also find that additional loss factors beyond top-down control of *Prochlorococcus* may be ecologically relevant, raising new questions on governing mechanisms in surface ocean ecosystems.

## Results

### Fitting ECLIP to field measurements of *Prochlorococcus*, cyanophages, and grazers

Time series data from the SCOPE HOE-Legacy 2A cruise (see Fig. 1b) reveals *Prochlorococcus* abundances are periodic, peaking at night and reaching their minima during the day[22]. Population abundances of heterotrophic nanoflagellates fluctuate with unclear periodicity[23], as do the abundances of T4- and T7-like cyanophage[22]. In contrast, the fraction of cells infected by T4- and T7-like viruses[22] are periodic, peaking at night. We recapitulated these periodicity analyses in Supplementary Note 1. This periodicity suggests the potential for diel-driven emergent synchronization in the food web, similar to community-wide metabolism in the NPSG[42].

To explore potential coexistence dynamics of *Prochlorococcus*, viruses, and grazers, we fit the Ecological Community driven by Light including Infection of Phytoplankton (ECLIP) model via MCMC given biologically realistic parameter bounds (see Fig. 1a for model schematic, "Methods" for model details, Table S3 for priors, Fig. S2 for division-associated priors, and the Supplementary Information for MCMC fitting details). The models are fitted against detrended empirical data, so for visualization we add this trend to the model simulations. The fitting of ECLIP with differing levels of grazer generalism are shown in Fig. 2. All ECLIP models were able to simultaneously reproduce the magnitudes of the different time series, producing fits with similar log-likelihoods (Fig. S9) while exhibiting statistical evidence of convergence (Figs. S5 and S8), even if it is not feasible to identify a particular, preferred level of grazer generalism. In sum, a range of nonlinear mathematical models including feedback between cyanobacteria, cyanophage, and grazers can jointly recapitulate multi-trophic population dynamics in the NPSG. Despite fitting overall magnitudes and oscillations in *Prochlorococcus* abundances, ECLIP underestimated the strength of oscillations in infected cells (an issue we return to later in the "Results").

### Interpretation of ecological mechanisms underlying model-data fits

The equivalence in model-data fits across a spectrum of grazer generalism suggests that differentiating model mechanisms requires inspection of posterior parameter fits. Posterior parameter values are shown in Fig. 3, with full details on division function fits in Fig. S3 and comparison with priors in Fig. S4. Most life-history traits converge to similar parameter regimes across ECLIP models, with a notable exception: a systematic trend in the grazer loss parameter $m_G$, reflecting a trade-off between grazer losses and gains via increasing $\gamma$. The inferred grazer loss rates correspond to grazer residence times of between 16.81 (95% CI: 15.37–20.19) days in the specialist model to 1.82 (95% CI: 1.8–1.87) days in the most generalist model when $\gamma = 0.5$ per day. These residence times are consistent with the range of estimated heterotrophic nanoflagellate doubling times in the Mediterranean Sea of 4–20 days[43]. Corresponding virus residence times were estimated as 1.16 days (95% CI: 0.59–5.3 days) in the specialist model and 1.17 days (95% CI: 0.57–8.32 days) when $\gamma = 0.5$ (see Fig. S10). MCMC posterior distributions appear tight (e.g., for $\mu_{ave}$, $\delta_t$, and $\phi$) or loose (e.g., for $\beta$, $m_P$, and $m_V$) suggesting differing parameter space sensitivities[44].

Figure 4 shows that $\gamma$ corresponds to a grazer specialism-generalism gradient in the inferred ECLIP models, with the relative contributions of *Prochlorococcus* consumption to grazer growth rates decreasing with increasing generalism, as expected (note life-history trait interdependence, as in equation 20, did not guarantee this result). However, absolute per-capita grazer consumption of *Prochlorococcus* only varied modestly ($\approx 0.04$–$0.075$ day$^{-1}$) between models. Hence, we interpret these findings to mean that per-capita grazing mortality of *Prochlorococcus* is relatively invariant to model choice and can be inferred robustly from model-data fits.

### Partitioning *Prochlorococcus* losses between top-down and other effects

We analyzed the predicted partitioning of *Prochlorococcus* mortality among grazing by heterotrophic nanoflagellates, viral-induced lysis by T4- and T7-like cyanophages, and other sources of mortality, using the inferred ECLIP models. We used posterior estimates from model-data fits to estimate total *Prochlorococcus* loss rates:

$$m_{total} = m_{lysis} + m_{grazing} + m_{other} \tag{1}$$

where: $m_{lysis} = \eta I$, $m_{grazing} = \psi(S + I)G$, $m_{other} = m_P(S+I)^2$ (see Supplementary Information for details). The proportion of each mortality

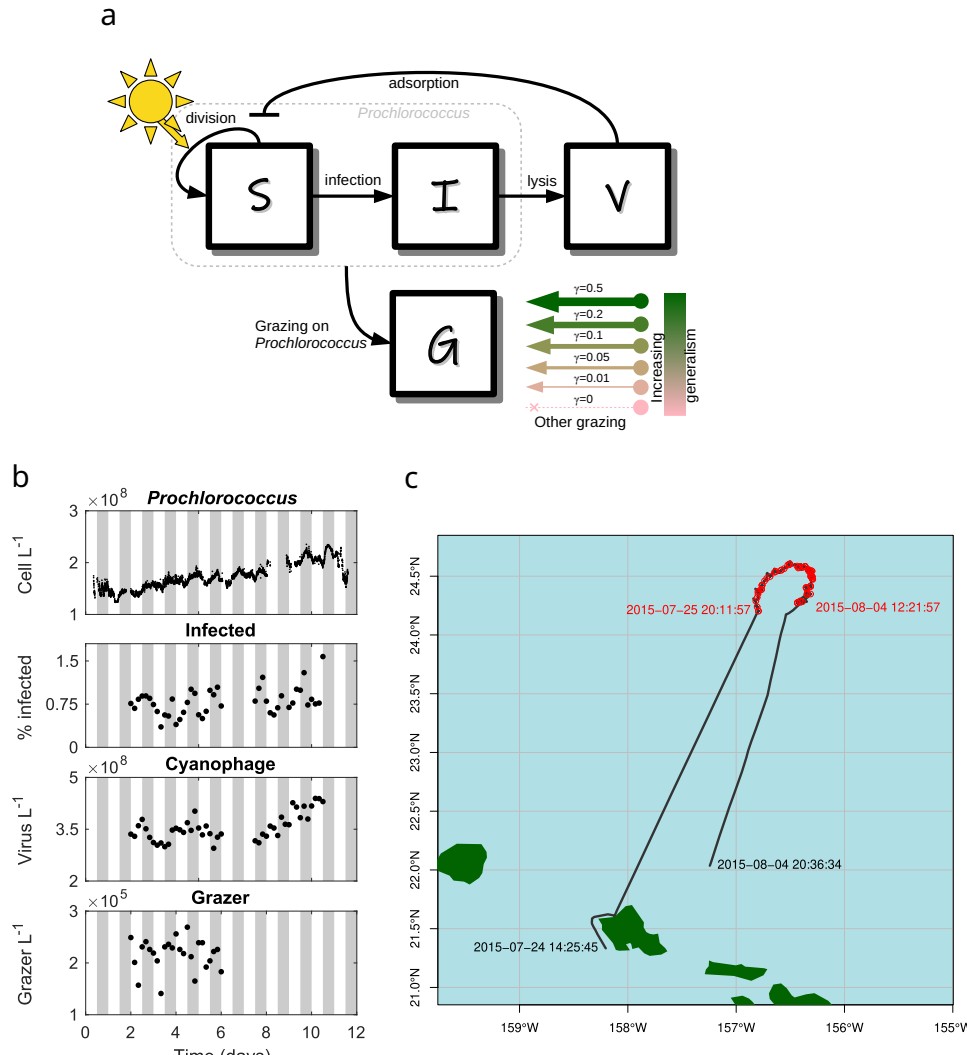

**Fig. 1 | Community ecological model of viral and grazer mediated predation; and SCOPE HOE-Legacy 2A cruise field data. a** *Prochlorococcus* are structured by infection status. Viruses (*V*) can infect susceptible *Prochlorococcus* cells (*S*) generating infected cells (*I*). Viral-induced lysis of infected cells releases virus particles back into the environment. Susceptible and infected *Prochlorococcus* cells are subject to grazing pressure from heterotrophic nanoflagellate grazers (*G*). Grazers may have a generalist strategy (e.g., grazing on heterotrophs, mixotrophs, and phytoplankton not represented by *S* and *I*). We specify six models along this specialism-generalism gradient by setting a parameter *γ*. When *γ* = 0 heterotrophic nanoflagellate grazers act as specialists and only consume *Prochlorococcus*; and as *γ* increases, *Prochlorococcus* constitutes less of the diet of heterotrophic nano-flagellate grazers. Parameters and units are specified in Table S2. **b** Reported empirical population dynamics of *Prochlorococcus* cells[74], the percentage of *Prochlorococcus* cells infected with T4/T7-like cyanophage[22], the abundance of free-living T4/T7-like cyanophage[22], and the abundance of heterotrophic nanoflagellate grazers[23]. **c** Cruise track and sampling stations. Local times (HST) for the start and end of recorded underway sampling (black line), and first and last sampling stations (red points) are annotated.

process is calculated as the average ratio of the component mortality rate relative to that of the total over the empirical time-series. Mortality partitioning suggests 87–89% (with extents of 95% confidence intervals ranging 66–96%) of *Prochlorococcus* losses were ascribed to grazing, 6% (with extents of 95% confidence intervals ranging 4–9%) to viral lysis and 4.5–6.6% (with extents of 95% confidence intervals ranging from less than 1% to 27%) to other mechanisms. The distribution of losses by category are shown in Fig. 5, with the top six rows in each panel denoting distinct grazer generalism levels, from *γ* = 0 to *γ* = 0.5. Inferred mortality estimates from ECLIP were relatively invariant regardless of the grazer generalism level. Hence, our estimates of mortality partitioning are robust to model choice, including the finding that grazer-induced losses predominate when jointly estimating the collective effects of grazers and viruses on multitrophic population dynamics. Notably, other forms of loss may be as

ecologically relevant as viral-induced mortality to daily *Prochlorococcus* losses.

## Contrasting *Prochlorococcus* loss estimates

ECLIP model-data integration simultaneously infers the putative daily loss of *Prochlorococcus* due to grazing, viral lysis, and other loss mechanisms. These joint estimates can be compared to alternative methods that estimate viral infection or grazing losses, albeit one factor at a time. Multiple approaches exist to infer *Prochlorococcus* loss rates in situ. For viral lysis, we consider two methods: (i) encounter theory; (ii) iPolony estimates. Conventional 'encounter' estimators use biophysical theory to estimate an upper-limit of size-dependent contact rates. However, encounter need not imply a successful adsorption and lysis event, hence the realized level of lysis is often significantly less than expected from biophysical limits[16]. In contrast, the iPolony

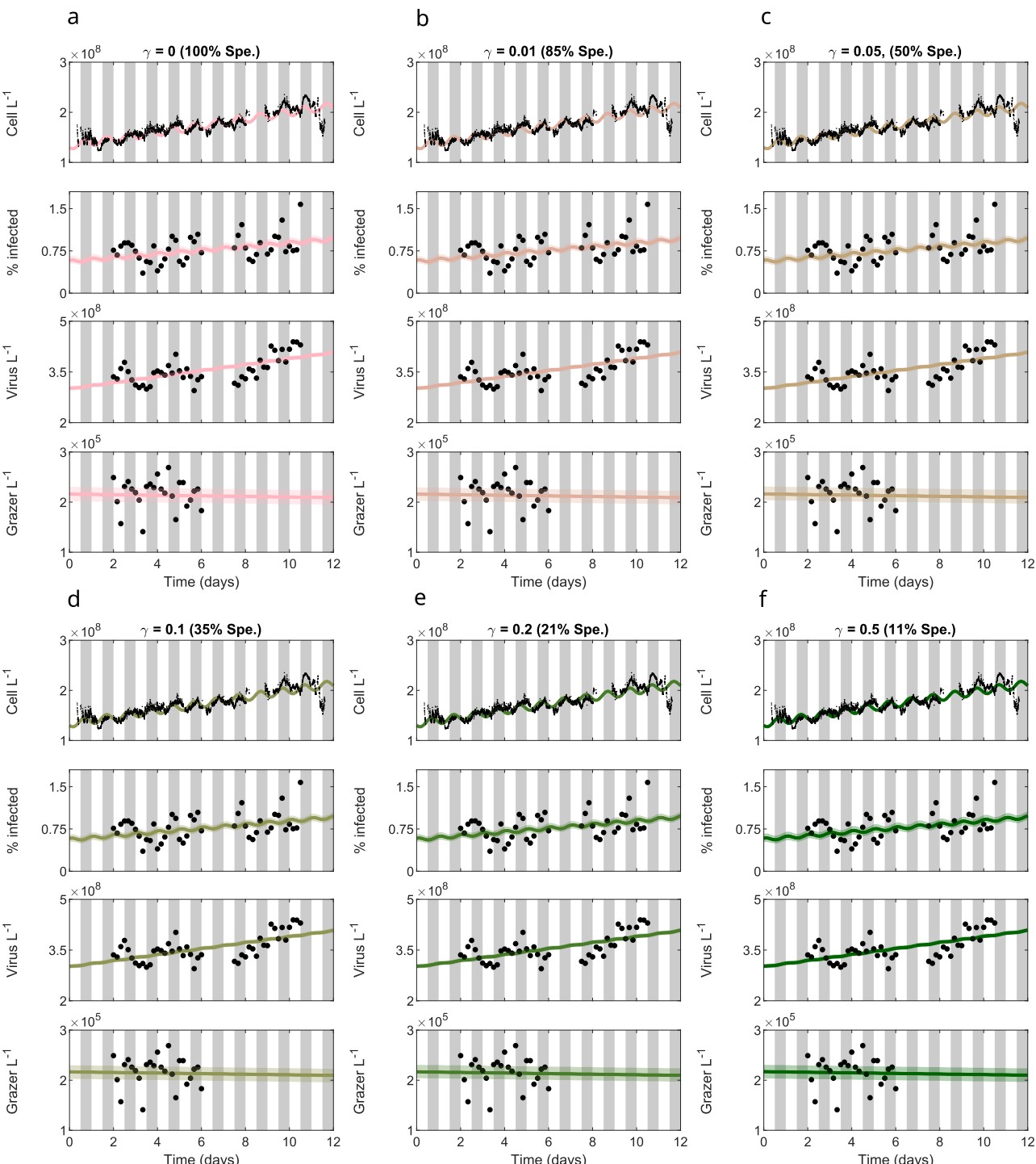

**Fig. 2 | Models across the specialism-generalism gradient fit empirical data.** ECLIP models (lines) are compared against empirical data (points). Model lines represent the median MCMC solution within 95% CI range found by the converged chains, shown as bands with colors representing the choice of $\gamma$. Data signals include *Prochlorococcus* cell abundances (top), the percentage of infected *Prochlorococcus* cells, the abundance of free viruses and the abundance of heterotrophic nanoflagellate grazers (bottom). The models were fitted against detrended data; for visualization we have added these trends to the model solutions. Gray bars indicate nighttime. Model solutions with: **a** $\gamma = 0$ (grazers act as specialists), **b** $\gamma = 0.01$, **c** $\gamma = 0.05$, **d** $\gamma = 0.1$, **e** $\gamma = 0.2$, **f** $\gamma = 0.5$ day$^{-1}$. The degree of grazer specialism (Spe.) is shown in parentheses above each subplot.

method quantifies the fraction of host cells infected by a target phage, which can be combined with estimates of viral latent periods and cell division rates to infer loss rates[22]. Regarding grazing, we consider three methods: (i) encounter theory; (ii) quota-based theory (as in ref. 23); (iii) flourescently-labeled bacteria (FLB) estimates. For grazers, size-dependent encounter rate theory and quota-based theories use biophysical contact rates and allometrically derived elemental growth requirements, respectively, to estimate grazing-induced loss rates. Theoretical estimates via encounter and quota methods have significant variability, leading to unconstrained mortality estimates, in part due to life-history trait uncertainties. FLB is a direct method, albeit relying on surrogate prey uptake as a proxy for cyanobacteria

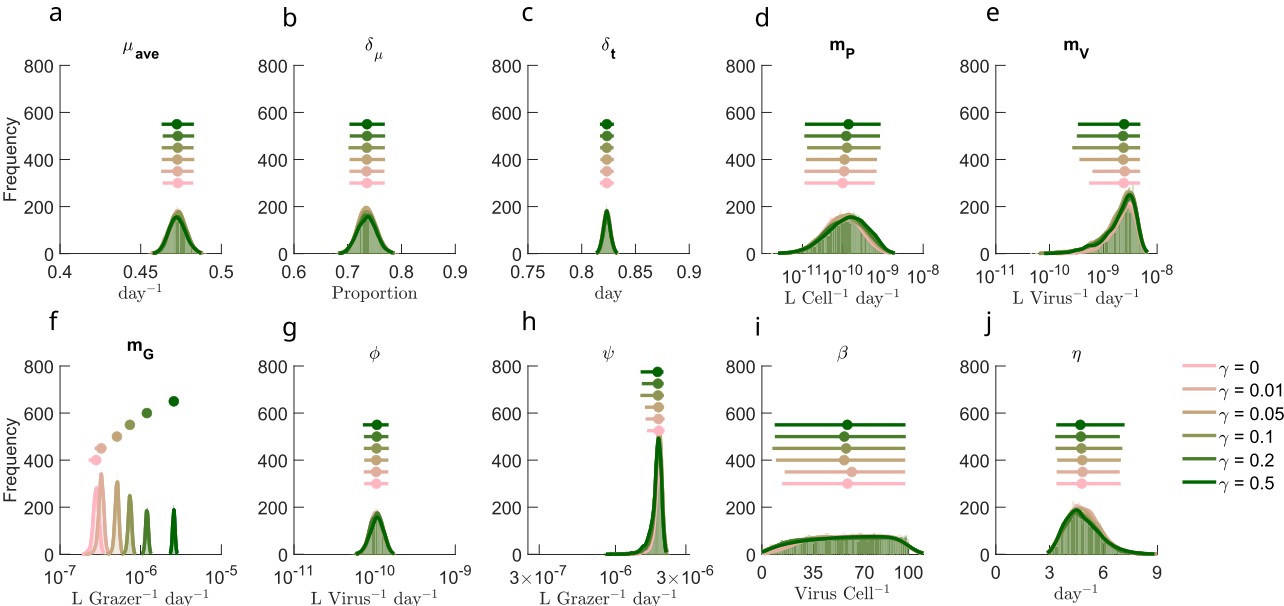

**Fig. 3 | Differences in inferred life-history traits across the specialist-generalist gradient. a–j** Parameter posterior distributions for different ECLIP models. Parameters are **a** $\mu_{ave}$: average *Prochlorococcus* division rate, **b** $\delta_\mu$: division rate amplitude, **c** $\delta_t$: phase of division rate, **d** $m_P$: higher order *Prochlorococcus* loss rate, **e** $m_G$: higher order viral loss rate, **f** $m_G$: higher order grazer loss rate, **g** $\phi$: viral adsorption rate, **h** $\psi$: grazer clearance rate, **i** $\beta$: viral burst size, and **j** $\eta$: viral-induced lysis rate. Jittered median (dot) and 95% CI range (horizontal line) for each of the models are shown above density plots. Full details of parameter bounds are shown in Table S2; see Supplementary Information for more details.

uptake rates (e.g., ref. 23 use *Dokdonia donghaensis* as surrogate prey). Figure 5 compares joint ECLIP-inferred relative mortality estimates with one-factor estimates from field-based iPolony measurements and fluorescently labelled bacterial (FLB) uptake estimates (for details of these and alternative, theoretical methods (encounter and quota), see Supplementary Note 2). For viral lysis, one-factor estimates using encounter theory do not constrain daily loss rates. For example, if lysis was limited by contact with host cells, then observed viral abundances could account for nearly 100% of observed *Prochlorococcus* loss. Conversely, estimated daily loss rates could decrease to nearly 0% if contact rates were significantly lower than biophysical limits suggest, adsorption was inefficient, or adsorption did not necessarily imply a successful infection because some phage were non-infective and/or defective. In contrast, quantitative estimates of infection processes in the NPSG via the iPolony method suggest viral-induced lysis by T4- and T7-like cyanophages contribute a comparatively small amount to *Prochlorococcus* cell losses (<5%). Likewise, accounting for observed grazer abundances and biophysically plausible grazing rates suggests grazing could explain daily *Prochlorococcus* losses alone. But, lower limits of theory based estimates of grazing-induced mortality—as was the case for viral-induced mortality—are poorly constrained given uncertainties in grazing efficiency. For example, empirically-derived FLB estimates suggest grazing by heterotrophic nanoflagellates accounts for up to ≈30% of total *Prochlorococcus* loss rates (although we note the original FLB experiment was designed to differentiate relative grazing by mixotrophic and heterotrophic nanoflagellates). If the iPolony and FLB methods were unbiased, then the majority of loss rates would be unaccounted for by top-down effects.

Instead, our mortality estimate comparison constrain the magnitude of distinct loss factors. Like the iPolony method, our joint ECLIP-inferred estimates suggest that viral-induced lysis is responsible for a small proportion of *Prochlorococcus* losses in the NPSG. We note that ECLIP-inferred viral lysis estimates are low but are ≈3% higher than those inferred from iPolony (see "Discussion"). In contrast, joint ECLIP-inferred estimates suggest grazing by heterotrophic nanoflagellates represents the majority of *Prochlorococcus* cell losses across ECLIP—far above that

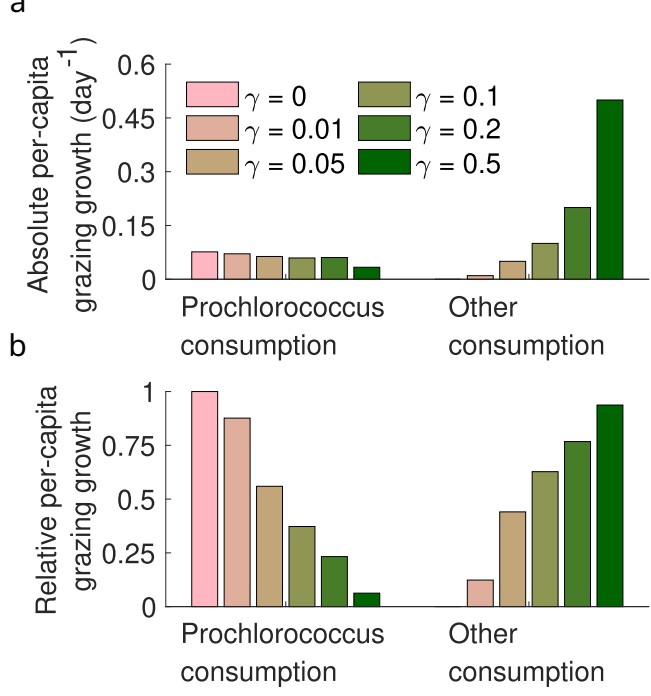

**Fig. 4 | Model differences across the specialist-generalist gradient. a, b** Inferred grazer growth attributable to consumption of *Prochlorococcus* or other sources (see Supplementary Information equation 20) across models.

inferred via FLB experiments. Notably, estimates of *Prochlorococcus* losses via grazing were robust to changes in grazer specialism, further suggesting FLB-derived estimates under-represent in situ grazing (which could represent biological or methodological uncertainty as hypothesized in ref. 23). In addition, the model-inferred combination of viral-induced lysis by T4- and T7-like cyanophages and grazing by

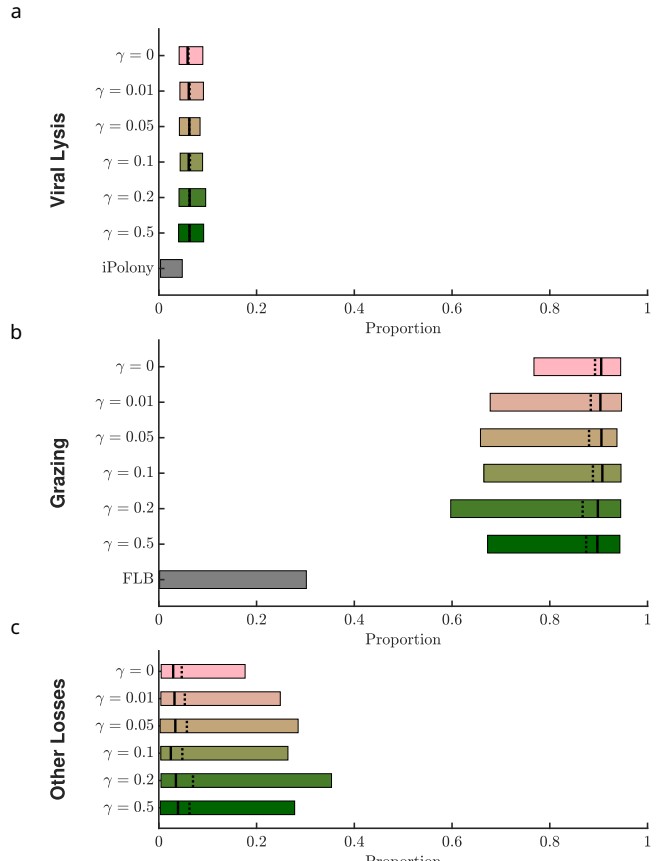

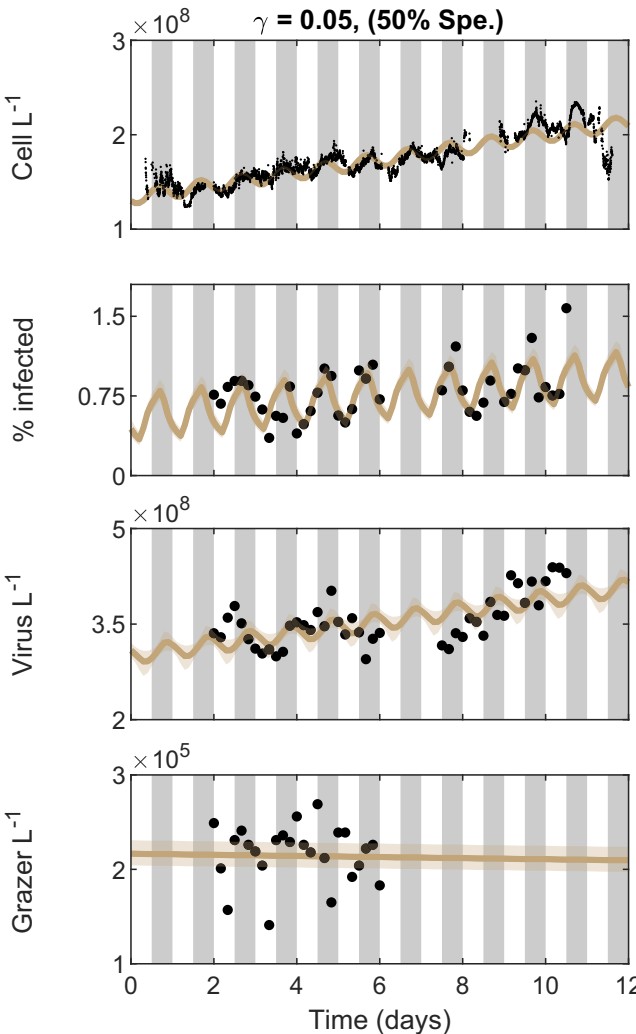

$\gamma$ = 0.05, (50% Spe.)

**Fig. 5 | Relative importance of viral lysis, grazing, and other effects on total *Prochlorococcus* mortality.** The proportion of mortality partitioned between **a** viral-induced lysis, **b** grazing, and **c** other sources for the ECLIP models and other measures of relative mortality. For ECLIP the results from all chains are shown. Bars in these panels denote mortality rate proportions associated with the 95% confidence intervals, where the mean and median are shown by solid and dashed lines, respectively. Other plotted measures of relative mortality are given via direct measurements of viral infection (iPolony), and Fluorescently Labelled Bacteria (FLB) incubation measurements (see Supplementary Note 2 for details).

**Fig. 6 | Diel-dependent adsorption rates improve fits to infected cells.** ECLIP model solutions with $\gamma = 0.05$ and diel-dependent adsorption rates are compared against empirical data in black. Model lines represent the median MCMC solution within 95% CI range found by the converged chains, shown as bands. Data signals include *Prochlorococcus* cell abundances (top), the percentage of infected *Prochlorococcus* cells, the abundance of free viruses and the abundance of heterotrophic nanoflagellate grazers (bottom). The models were fitted against detrended data; for visualization we have added these trends to the model solutions. Gray bars indicate nighttime. The degree of grazer specialism (Spe.) is shown in parentheses above the plot.

heterotrophic nanoflagellates typically does not sum to equal 100% of total *Prochlorococcus* daily cell losses. Instead, model-data integration suggests other sources of *Prochlorococcus* cell loss account for ≈6% of daily losses (with 95% confidence intervals ranging from less than 1% to 27%). Together, both model-data fits and independent estimates of top-down mortality suggest other loss processes beyond grazing by heterotrophic nanoflagellates and lysis by T4/T7-like cyanophages may be ecologically relevant in shaping daily phytoplankton rhythms.

**Capturing diel periodicity of infected *Prochlorococcus***
The multi-trophic ECLIP model incorporating light-driven photosynthesis resolved the diel periodicity of total cell counts of *Prochlorococcus* across a gradient of grazer consumption strategies (see Fig. 2). However, the ECLIP model did not recapitulate the magnitude of the observed periodicity of infected cells (as shown in Supplementary Note 1). One potential reason for this gap is that we did not incorporate the potential for plasticity in viral traits into ECLIP, in contrast to previous work which shows that cyanophage exhibit light-dependent adsorption rates to *Prochlorococcus*[21,45]. Diel-dependent adsorption may reflect changes in both cell physiology and cell size as *Prochlorococcus* cells grow during the day in G1 phase before synthesising DNA in S phase and transitioning to G2 phase in preparation to divide at night–larger cells are expected to have larger rates of adsorption[16] and darkness can modulate and arrest transitions through

cell cycle phases[1,46,47] which in turn could modulate viral infection[48]. Hence, we modified the core ECLIP model to include a time-dependent step-wise adsorption rate such that adsorption varies from 50% lower at dawn (midnight to noon) to 50% higher at dusk (noon to midnight) relative to the initially inferred adsorption rates (preserving the mean adsorption rates used in Fig. 2). The emergent community dynamics preserve the timing and magnitude of oscillations in *Prochlorococcus* populations while also inducing oscillations in infected cells (see Fig. 6 for the case with $\gamma = 0.05$ and Fig. S11 for the full suite of grazer strategies). Hence, we find that it is possible to recapitulate the daily community time series insofar as we incorporate both light-dependent cellular and viral traits.

**Sensitivity of the magnitude and source of *Prochlorococcus* mortality to parameter variation**
We incorporate a parameter sensitivity analysis to address the robustness of inferred top-down loss rates and source of mortality for

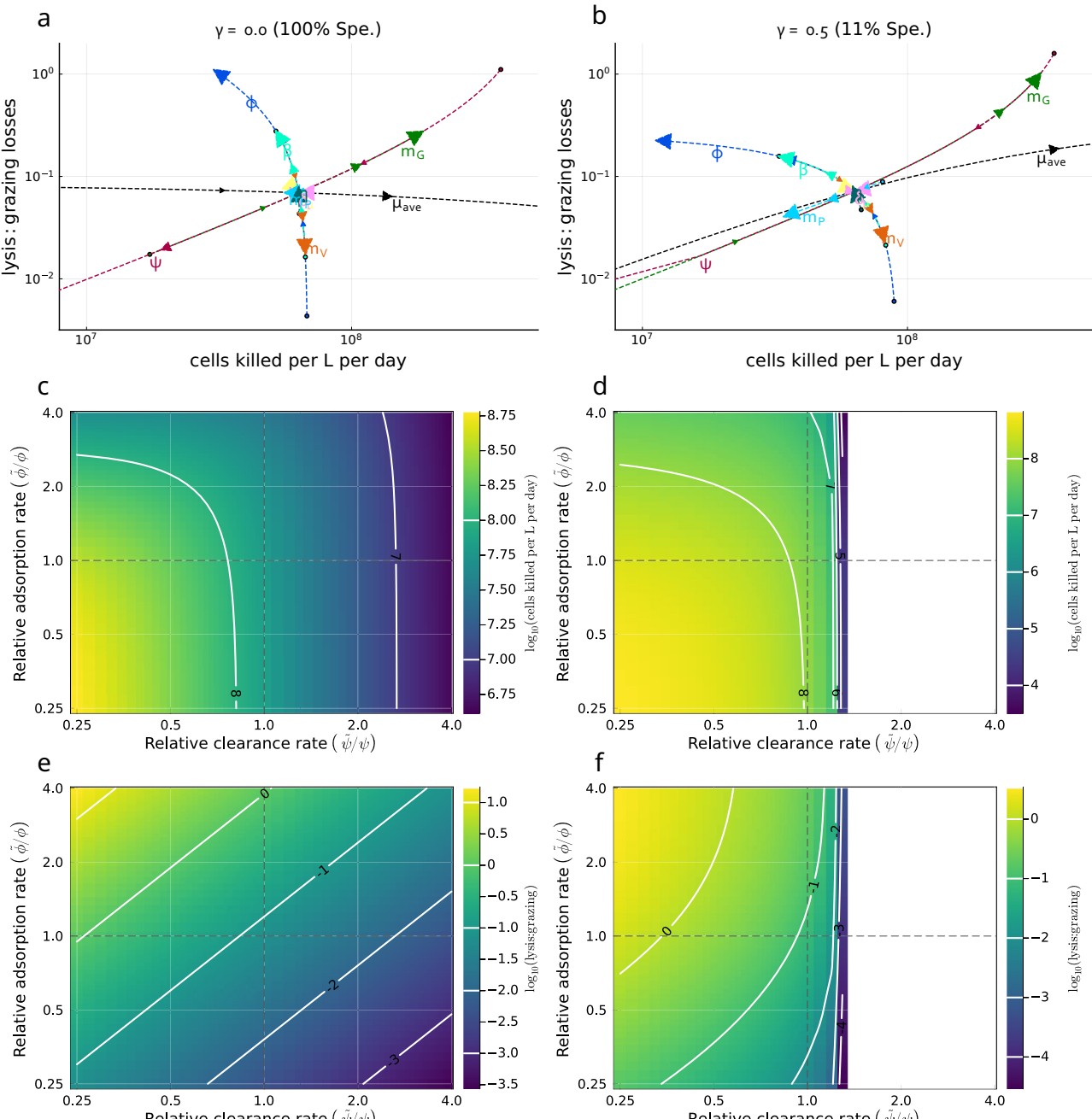

**Fig. 7 | Assessing robustness of the estimated magnitude and source of *Prochlorococcus* daily mortality.** Sensitivity analysis was conducted based on the MCMC inferred parameter sets for the detrended model with specialist grazing ($\gamma = 0$, 100% Spe.) (**a**, **c**, **e**) and the most generalist grazing model ($\gamma = 0.5$, 11% Spe.) (**b**, **d**, **f**). **a**, **b** A single parameter is varied at a time with all others fixed to evaluate changes in *Prochlorococcus* daily mortality and the relative role of viral-induced lysis vs. grazing. Arrows, and label positioning relative to the intersection, indicate the effect of increasing each parameter, with circles denoting the smallest value of each parameter. Label and line colors are the same for each parameter varied. Parameters were varied from 0.25x to 4x the baseline value. Parameters are $\mu_{ave}$: average *Prochlorococcus* division rate, $\delta_\mu$: division rate amplitude, $\delta_t$: phase of

division rate, $m_P$: higher order *Prochlorococcus* loss rate, $m_G$: higher order viral loss rate, $m_G$: higher order grazer loss rate, $\phi$: viral adsorption rate, $\psi$: grazer clearance rate, $\beta$: viral burst size, and $\eta$: viral-induced lysis rate. **c**–**f** Covariation between adsorption ($\tilde{\phi}$) and clearance rates ($\tilde{\psi}$) relative to their MCMC inferred values (respectively, $\phi$ and $\psi$) and the effect on *Prochlorococcus* daily mortality (**c**, **d**) and the lysis:grazing ratio (**e**, **f**). Dashed lines indicate the MCMC inferred value. Contours in (**c**–**f**) represent differences in magnitude. Note $10^0$ in (**e**, **f**) represents the case when grazing losses are equal to viral-induced losses ($10^0 = 1$). White space regions in (**d**, **f**) denote scenarios when *Prochlorococcus* abundance becomes less than 1 cell per L.

*Prochlorococcus* as inferred via MCMC. In Fig. 7 we evaluate parameter sensitivity using baseline MCMC model parameters as inferred for the specialist ($\gamma = 0$) condition (left panels, a, c, e) and the generalist ($\gamma = 0.5$) condition (right panels, b, d, f). First, we individually varied each parameter between $\frac{1}{4}$ and 4 times the baseline value, while keeping all other parameters fixed; and assessed the detrended model

dynamics after a simulation of 1000 days to avoid a transient and reach stationary dynamics. We find differences in the degree and direction to which parameter variation affects realized mortality rates. In particular, we find that there are three 'arcs' (Fig. 7a, b) in which the model is most sensitive—(i) growth arc: increasing the growth rate $\mu$ increases cell losses per day without substantively altering the relative losses

caused by grazing or viral-induced lysis, (ii) viral arc: increasing the adsorption rate $\phi$, increasing the burst size $\beta$, or decreasing the viral loss term $m_V$ reduced overall *Prochlorococcus* cell losses per day, while increasing the ratio of lysis:grazing induced loss; (iii) grazer arc: increasing the grazing clearance rate $\psi$ or reducing the grazing loss term $m_G$ reduced overall *Prochlorococcus* cell loss per day, while also reducing the ratio of lysis:grazing induced loss. Notably, given local parameter variation along the grazer (Fig. 7a, b) or viral arc (Fig. 7a), lysis-induced loss rates could become as or more important than grazing induced loss (i.e., exceeding 1). We also observe that incorporating grazer generalism preserves the qualitative outcomes while shifting quantitative sensitivity (as well as transforming coexistence in certain parameter regimes). In particular, when nanoflagellate grazers consume additional prey sources (right hand side with $\gamma = 0.5$), grazers are able to sustain a larger baseline population and exert a greater baseline grazing pressure on *Prochlorococcus*. We observe that increases in grazing pressure beyond a critical point can lead to elimination of *Prochlorococcus*, grazers, and viruses (white region in panels Fig. 7d, f). Overall, the most sensitive model parameters included average growth rate, $\mu_{ave}$, grazing clearance rate, $\psi$, grazer loss rate, $m_G$, and the adsorption rate, $\phi$ (see Fig. 7a, b and Table S4). We chose to additionally assess model sensitivity by co-varying $\psi$ and $\phi$ (Figs. 7c, d & 7e, f). We find a wide region of parameter space with realistic daily *Prochlorococcus* cell losses—near to $10^8$/L/day compatible with higher and lower clearance and adsorption life-history traits. We also find higher daily *Prochlorococcus* cell losses when both adsorption and clearance rates are reduced where niche competition (denoted as higher order-losses in equation 1: $m_S S(S + I) + m_J J(S + I)$) becomes a more important mortality term. Finally, we find that the estimated level of lysis:grazing given parameter variation is compatible with increasing importance of viral-induced lysis rates. Typically, higher viral adsorption and lower grazer clearance rates lead to an increasing importance of viral lysis relative to grazing. This parameter sensitivity analysis reinforces the finding that viral-induced lysis rates may represent a relatively low fraction of mortality in the NPSG for *Prochlorococcus* and that changes in predator and viral loss rates can lead to circumstances where viral lysis exceeds 20% of total mortality, or even exceeds grazer-induced lysis altogether.

## Discussion

We developed and analyzed a multitrophic community ecology model (ECLIP) consisting of *Prochlorococcus*, viruses, and grazers to investigate feedback mechanisms and ecological drivers of oligotrophic ocean microbial population dynamics on diel timescales. ECLIP can recapitulate the dynamical coexistence of cyanobacteria, viruses infecting cyanobacteria, and grazers population abundances in the NPSG. By combining model-data fits with direct measurements of mechanistic interactions we infer that grazing rather than viral-induced lysis predominates in shaping *Prochlorococcus* mortality in NPSG surface waters. We also find that the combination of lysis and grazing does not fully account for daily *Prochlorococcus* losses. Instead, model-inference suggests the ecological relevance of other density-dependence loss mechanisms for *Prochlorococcus* in the NPSG.

Overall, model-data fitting to NPSG measurements enabled us to examine how much *Prochlorococcus* mortality can be ascribed to viral lysis, heterotrophic nanoflagellate grazing, or other loss processes. In partitioning *Prochlorococcus* mortality, we found different outcomes across model scenarios and independent auxiliary estimates (Fig. 5). Indirect estimates via encounter or quota-based theory are poorly constrained and limited by our current knowledge of ecological life-history traits. However, fitting ECLIP to field data resulted in more constrained mortality estimates. We infer that viral-induced mortality of *Prochlorococcus* in this system is relatively weak, consistent with prior estimates[22]. Low levels of viral-induced *Prochlorococcus* (and *Synechococcus*) mortality have also been found in the Sargasso Sea[49],

and the Mediterranean and Red Sea[50]. This could be a defining characteristic of viral impacts on cyanobacteria in oligotrophic gyres—as opposed to more dynamic ocean regions where viral mortality can be considerably more substantial[51,52]. Indeed, life-history trait variation model sensitivity analysis (Fig. 7) supports the potential for viral-induced lysis to rival grazing mortality. Field-based campaigns that concurrently measure grazing and viral impacts will help further constrain the balance between viral- and grazing-induced mortality.

Direct mortality estimates from grazing incubation experiments and infected cell measurements also provide evidence that heterotrophic nanoflagellate grazing and T4- and T7-like viral-induced mortality do not account for all *Prochlorococcus* losses in the NPSG. Quantifying the relative importance of mortality processes beyond conventional top-down effects (grazing and lysis) is critical for understanding how grazers and viruses contribute to mortality and energy transfer in marine microbial communities[2,24,25,29]. Interestingly, our ECLIP analysis suggested higher levels of grazing mortality than from FLB measurements. This suggests grazers do not uptake this biological tracer at the same rate as *Prochlorococcus*—potentially reflecting differences in chemical composition, size, or experimental conditions[23].

The finding that model-data fits impute other sources of mortality as quantitatively significant suggests other feedback mechanisms should be included in model representations of marine surface community dynamics. A missing component in our modeling framework is the effects of mixotrophic nanoflagellates[23,53–56] which are likely the main source of additional losses. We did not include mixotrophs in our framework due to experimental difficulties in separating phototrophic from mixotrophic nanoflagellates, and theoretical challenges of appropriate physiological modeling. However, surface ocean phytoplankton losses plausibly include factors beyond grazing and viral-induced lysis[57,58]. In Supplementary Note 4 we review potential mechanisms contributing to the unaccounted losses of *Prochlorococcus*, beyond those from heterotrophic nanoflagellate grazing and T4- and T7-like viral-induced lysis. These include ecological feedbacks leading to distinct functional and/or light-driven responses, aggregation and/or sinking, stress, population heterogeneity, and the possibility of having missed other top-down mortality. Similarly, our measurements may miss population heterogeneities within *Prochlorococcus* masking our ability to interpret average per-capita mortality. Investigating alternative mechanisms of *Prochlorococcus* losses may improve understanding of how biomass and nutrients are transferred through marine food webs. Further investigation and characterization of growth dynamics may also be warranted, as mischaracterization may impact our ability to infer mortality processes—note, at depth *Prochlorococcus* has recently been shown to rely on mixotrophic strategies[59].

The ECLIP framework comes with caveats, despite inclusion of multiple populations and interactions. First, we focused on the impacts of direct, light-driven forcing of cyanobacterial division—hence, oscillations arising in other components reflect a combination of instabilities that can arise in nonlinear population models as well as the cascading impacts of such oscillations on the community. Unlike other picoplankton modeling efforts including generic loss terms, e.g., refs. 2,10,12,47, ECLIP can infer loss partitioning between grazing, lysis, and other processes. However, ECLIP does not explicitly capture size-structured processes which are important drivers of growth[2,10,12,47] and other ecological interactions[16]. In addition, light-driven forcing of division does not fully account for variability in processes such as nutrient content[42,60,61], and metabolic state[42]. While these attributes are not specifically modeled, they may have bearing on inferring life-history traits. Further, we note that (co)evolutionary dynamics within microbial systems, especially with respect to the viruses, can occur on rapid timescales[62], which has the potential for strain-level differentiation in life-history traits and infection networks which can alter and

potentially control population dynamics[63–66]. ECLIP provides a complementary framework for understanding marine microbial ecology; and we hope future efforts will attempt to blend these types of models. Direct incorporation of diel impacts on grazing[18,20,23,67] or other viral traits (e.g., beyond adsorption)[21,68] may be required to mechanistically understand population dynamics on sub-daily timescales—and particularly the magnitude of oscillations, including infected cell abundances. Second, we have used two focal processes to examine how carbon and other nutrients in basal picoplankton are transferred, either up the food chain via grazing, or retained in the microbial loop via viral lysis (aka the viral shunt)[33,34]. This dichotomy reflects potential tension regarding the extent to which primary production stimulates the biological pump requiring further investigation. Model extensions could include mechanistic process of export explicitly, whether through coupling grazing to the generation of particles and/or examining the extent to which viral lysis generates aggregates which can be exported to the deep oceans via the viral shuttle[36,69]. Finally, our work has identified a potential accounting challenge in quantifying the balance of *Prochlorococcus* growth and losses. Despite the daily growth and division of cells, overall abundances remain tightly constrained—our work suggests this constraint depends on factors beyond loss ascribed to T4- and T7-like cyanophage and nanoflagellate grazers.

In summary, the ECLIP multitrophic modeling framework provides opportunities to disentangle putative mechanisms underlying the control of microbial surface ocean populations. The model provides support for the dominant role of grazers in controlling *Prochlorococcus* in the NPSG, that relatively high viral abundances can be compatible with relatively low infection (and viral-induced mortality) rates, and that the relative balance of loss due to grazing vs. lysis is context-dependent. The analysis also identifies a key direction for future work: identifying the potentially 'missing mortality' at the base of the marine food web. Moving forward, in situ observations are needed to probe aggregation and sinking, autolysis, programmed cell death, or other forms of loss of *Prochlorococcus* and to understand the feedbacks of coupled variation in cyanobacterial growth and loss in a changing ocean.

## Methods
### Model overview
We developed a mechanistic mathematical model of an Ecological Community driven by Light including Infection of Phytoplankton (ECLIP). The mechanistic model is driven by parameterized interactions with ecological interpretations. Our model includes dynamics of *Prochlorococcus*, grazers, and viruses, as well as *Prochlorococcus* division and loss, where the loss arises due to a combination of grazing, viral lysis, and other factors (see Fig. 1a). In this model, viruses correspond to the abundances of T4- and T7-like cyanophages, known to primarily infect *Prochlorococcus*. Grazers represent heterotrophic nanoflagellates which feed on multiple prey types[70], however, the primary prey for heterotrophic nanoflagellates could be *Prochlorococcus*[23]. We introduced flexibility in our framework to account for the extent to which *Prochlorococcus* constitutes the primary food source for the observed grazer class. To assess this uncertainty we investigated six grazer strategies, ranging from a "specialized" grazer class exclusively consuming *Prochlorococcus* cells ($\gamma = 0$ day⁻¹) to models with increasing levels of generalism ($\gamma = 0.01$ to $\gamma = 0.5$ day⁻¹) representative of grazers consuming additional prey, e.g., heterotrophic bacteria which are not explicitly integrated into the model. Mixotrophic nanoflagellates[71] were observed, but contribute less to the grazing pressure on the bacterial community compared to heterotrophic nanoflagellates[23]. As it was not possible to differentiate abundance measurements of mixotrophic nanoflagellates from phototrophic nanoflagellates[23], we focus only on grazing by heterotrophic nanoflagellates. Across the gradient of specialism-generalism grazer

models, we searched for biologically feasible parameters using a model-data integration approach to generate dynamics consistent with observed population dynamics in the NPSG.

### Ecological model of phytoplankton communities with viral and grazer mediated predation (ECLIP)
The ECLIP model represents *Prochlorococcus* cell division as a light-driven process where cell division is expected to occur at night[2,72] and *Prochlorococcus* cell losses are controlled by viral lysis, grazing, and other density-dependent factors (Fig. 1a). The *Prochlorococcus* population is divided into cells that are susceptible to viral infection ($S$) and cells that are infected ($I$) by viruses ($V$). Grazers ($G$) feed indiscriminately on both $S$ and $I$ classes. Abundance dynamics of $S$, $I$, $V$, and $G$ over time are described by the following equations:

$$\frac{dS}{dt} = \overbrace{\mu(t)S}^{division} - \overbrace{m_P S(S+I)}^{higher-order\ losses} - \overbrace{\phi SV}^{infection} - \overbrace{\psi SG}^{grazing}$$

$$\frac{dI}{dt} = \overbrace{\phi SV}^{infected} - \overbrace{m_P I(S+I)}^{higher-order\ losses} - \overbrace{\eta I}^{viral-lysis} - \overbrace{\psi IG}^{grazing}$$

$$\frac{dV}{dt} = \overbrace{\beta \eta I}^{viral\ production} - \overbrace{\phi(S+I)V}^{adsorption} - \overbrace{m_V V^2}^{higher-order\ losses} \quad (2)$$

$$\frac{dG}{dt} = \overbrace{\epsilon \frac{N_P}{N_G} \psi(S+I)G}^{grazing\ on\ Prochlorococcus} + \overbrace{\gamma G}^{generalist\ grazing} - \overbrace{m_G G^2}^{higher-order\ losses},$$

where

$$\mu(t) = \mu_{ave}(1 + \delta_\mu \sin(2\pi(t + \delta_t))). \quad (3)$$

*Prochlorococcus* have a diel-driven division rate $\mu(t)$ whose proportional amplitude and phase are set by parameters $\delta_\mu$ and $\delta_t$, respectively, and $t = 0$ represents 06:00:00 local time ($t$ in days) (see Fig. S1). *Prochlorococcus* have a nonlinear loss rate, $m_P$, dependent on total *Prochlorococcus* abundance, implicitly representing niche competition[73]. Viruses infect susceptible *Prochlorococcus* at rate $\phi$. For each infection a burst size of $\beta$ new virions are released into the environment upon cellular lysis following the latent period (average duration $\frac{1}{\eta}$). Grazing upon *Prochlorococcus*, at rate $\psi$, is non-preferential regarding infection status. Consumed *Prochlorococcus* biomass is converted into grazer biomass with Gross Growth Efficiency (GGE) $\epsilon$ and assumed proportional to a nitrogen currency, given the nitrogen content in a *Prochlorococcus* cell ($N_P$) and a grazer ($N_G$), leading to an effective GGE of $\epsilon \frac{N_P}{N_G}$. We introduce $\gamma$ to denote the level of generalism in grazing, where $\gamma$ represents net additional gains to the grazer from non-*Prochlorococcus* prey sources after accounting for respiratory costs. A specialist strategy has $\gamma = 0$ day⁻¹, assuming that *Prochlorococcus* cells are the only grazer prey source. In contrast, we represent five generalist strategies via $\gamma = 0.01$ (very low), 0.05 (low), 0.1 (medium), 0.2 (high), or 0.5 (very high) day⁻¹, implying that grazers have a net positive growth rate even in the absence of $S$ or $I$ via consumption of other phytoplankton, heterotrophic bacteria, or grazers (intraguild predation). The degree that grazers act as generalists, rather than specialists on *Prochlorococcus*, depends on other life-history traits (Supplementary Information equation 20). Grazer and viral losses are characterized by nonlinear loss terms (with rates $m_G$ and $m_V$) to avoid structurally biasing the model to favor one *Prochlorococcus* predator type[29] and to avoid competitive exclusion between grazers and viruses. Linear loss terms were excluded to reduce the number of free parameters to estimate via inference. See further details in the Supplementary Information (see Table S2 for parameter definitions and Table S3 for specification of parameter priors).

## In situ measurements

We aggregate previously reported data collected from the Summer 2015 SCOPE HOE-Legacy 2A cruise (Fig. 1b, c)[22,23,74]. Measurements of total *Prochlorococcus* abundance were made every ≈ 3 min, with measurements of infected cells (infected by either T4- or T7-like cyanophages), total virus particles of either T4- or T7-like cyanophages, and heterotrophic nanoflagellate grazers collected at 4 h intervals over a multi-day period aboard the R/V Kilo Moana. In all figures, the 12 days shown represent 06:00:00 24 July 2015 to 06:00:00 5 August 2015 local time.

***Prochlorococcus* cell abundance.** SeaFlow−a shipboard in situ flow cytometer−continuously measures forward scattering, red and orange fluorescence intensities of particles ranging from ~0.4 to 4 μm in diameter from underway samples (continuously pumped surface seawater from ~7 m depth) every 3 min. A combination of manual gating and statistical methods was used to identify *Prochlorococcus* based on forward scatter (457/50 bandpass filter), orange fluorescence (572/28 bandpass filter) and red fluorescence (692/40 band-pass filter) relative to 1-μm calibration beads (Invitrogen F8823). Individual cell diameters were estimated from SeaFlow-based light scatter by applying Mie light scatter theory to a simplified optical model, using an refractive index of 1.38[74–76]. Data were obtained via Simons CMAP[77].

**Virus abundance and infection.** Samples for virus abundance and infection were collected every 4 h at 15 m depth using a CTD-rosette equipped with 12 L niskin bottles as described in ref. 22. Samples for cyanophage abundances (40 mL) were filtered through a 0.2 μm syringe top filter, flash frozen, and stored at −80 °C. Samples for infected cells (40 mL) were filtered through a 20 μm nylon mesh, fixed with electron microscopy grade glutaraldehyde (0.125% final concentration), incubated for 30 minutes in the dark at 4 °C, flash frozen, and stored at −80 °C. Cyanophage concentrations were analyzed using the polony method for T7-like[78] or T4-like[79] cyanophage families. Virally infected *Prochlorococcus* was quantified using the iPolony method[22] in which *Prochlorococcus* cells were sorted with a BD Influx cytometer and screened for intracellular T4-like and T7-like cyanophage DNA.

**Heterotrophic nanoflagellates.** Samples for nanoplankton (protists 2–20 μm in diameter) abundances were collected every 4 hours at 15 m depth[23]. Subsamples were preserved with formaldehyde (final concentration 1%, final volume 100 mL) and stored at 4 °C. Slides were prepared from preserved samples within 12 h of sampling by filtering 100 mL subsamples down to ~1 mL onto blackened 2 μm, 25 mm polycarbonate filters and staining the samples with 50 μL of a 4'-6'diamidino-2-pheylindole (DAPI, Sigma-Aldrich, St. Louis, MO) working solution (1 mg mL$^{-1}$) for 5–10 min in the dark[80]. Stained samples were filtered and rinsed; filters were placed on glass slides with a drop of immersion oil and coverslip, then sealed with clear nail polish. Slides were stored at −20 °C. Heterotrophic nanoplankton abundances were counted using epifluorescence microscopy from triplicate slides, and differentiated from photo/mixotrophic nanoplankton by the lack of chlorophyll *a* autofluorescence in plastidic structures when viewed under blue-light excitation[23].

## Model-data integration

To constrain ECLIP models to data, we used Markov Chain Monte Carlo (MCMC), implemented in the Turing package[81] in Julia[82]. MCMC is a class of Bayesian inference algorithms that aims to infer model parameter probability distributions given the structure of model equations, data, and prior parameter distributions. We used the No-U-Turn Sampler, a Hamiltonian Monte Carlo algorithm which does not require manual tuning and stops upon backtracking, to sample posterior distributions[83]. Further details are in the Supplementary Information.

## Reporting summary

Further information on research design is available in the Nature Portfolio Reporting Summary linked to this article.

## Data availability

The data sources that we use in our analysis have been published. *Prochlorococcus* cell abundances are from Ribalet et al. 2020, https://doi.org/10.5281/zenodo.3994953[74], free virus and percentage of infected *Prochlorococcus* cells are from Mruwat et al. 2021, https://doi.org/10.1038/s41396-020-00752-6[22], and heterotrophic nanoflagellate abundances are from Connell et al. 2020, https://doi.org/10.3354/ame01950[23]. All data from analyses, including for ECLIP and Bayesian parameter inference are archived on Zenodo, https://doi.org/10.5281/zenodo.10530398[84].

## Code availability

Code for ECLIP and performing Bayesian parameter inference was written and run in Julia, with some analyses and plotting performed in MATLAB and R. Core code for running model simulations, analysis and plotting is archived on Zenodo: https://doi.org/10.5281/zenodo.10530398[84]. Software used in our analyses and plotting are: MATLAB 9.12.0.1927505 (R2022a) Update 1, and the Econometrics ToolBox; Julia v1.6.3 and packages CSV v0.10.3, DataFrames v1.3.2, Distributions v0.25.49, Measures v0.3.1, LaTeXStrings v1.3.0, CairoMakie v0.7.4., MCMCChains v5.1.0, Plots v1.27.0, StatsPlots v0.14.33, DifferentialEquations v7.1.0, Turing v0.20.4, MCMCChains v5.1.0; R version 4.2.1 (2022-06-23) and packages rnaturalearth 0.1.0, vroom 1.5.7, sf 1.0-8, lubridate 1.8.0, lomb 2.1.0, RAIN 1.34.0 (from Bioconductor).

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

## Acknowledgements

We thank the captain and crew of the R/V Kilo Moana for their effort and assistance on the Hawaii Ocean Experiment Legacy II cruise (KM1513); we thank Michael Follows for feedback that helped improve the manuscript, and we thank Jeremy Harris for code review. This work was supported by grants from the Simons Foundation (no. 549894 to J.R.C., no. 827829 to C.L.F., no. 574495 to F.R., no. 721231 to J.S.W., and no. 329108 to E.V.A., D.A.C., D.L., A.E.W., and J.S.W). This is a contribution of the Simons Collaboration on Ocean Processes and Ecology (SCOPE). This work was produced under the auspices of the U.S. Department of Energy by Lawrence Livermore National Laboratory under Contract DE-AC52-07NA27344. LLNL-JRNL-859343. This research was supported in part through research cyberinfrastructure resources and services provided by the Partnership for an Advanced Computing Environment (PACE) at the Georgia Institute of Technology, Atlanta, Georgia, USA.

## Author contributions

S.J.B., D.D. and J.S.W. designed the study, with contributions from A.R.C., J.R.C., M.D., C.L.F., P.C., M.C.G.C., S.K.H., S.T.W., D.M., R.A.R.-G., S.P., K.W.B., D.R.M., E.V.A., D.A.C., D.L., A.E.W., and F.R. J.R.C., P.C., M.C.G.C., S.T.W., D.C., D.L. and F.R. contributed to data collection, sample processing, and data preparation. S.T.W. served as chief scientist for the research expedition. S.J.B. and D.D. led coding and data analysis along with A.R.C., R.A.R.-G., S.P., F.R. and J.S.W. All other authors (J.R.C., M.D., C.L.F., P.C., M.C.G.C., S.K.H., S.T.W., D.M., K.W.B., D.R.M., E.V.A., D.A.C., D.L., A.E.W.) contributed to interpretation of data analysis. S.J.B., D.D., and J.S.W. led the writing of the manuscript with contributions from A.R.C., J.R.C., M.D., C.L.F., P.C., M.C.G.C., S.K.H., S.T.W., D.M., R.A.R.-G., S.P., K.W.B., D.R.M., E.V.A., D.A.C., D.L., A.E.W., and F.R. All authors approved the final manuscript.

## Competing interests

The authors declare no competing interests.

## Additional information

[1]School of Biological Sciences, Georgia Institute of Technology, Atlanta, GA, USA. [2]Department of Biology, University of Maryland, College Park, MD, USA. [3]Sorbonne Université, CNRS, USR 3579, Laboratoire de Biodiversité et Biotechnologies Microbiennes (LBBM), Observatoire Océanologique, Banyuls-sur-Mer, France. [4]School of Physics, Georgia Institute of Technology, Atlanta, GA, USA. [5]Daniel K. Inouye Center for Microbial Oceanography: Research and Education, University of Hawai'i at Mānoa, Honolulu, HI, USA. [6]Department of Oceanography, University of Hawai'i at Mānoa, Honolulu, HI, USA. [7]Department of Earth, Atmospheric, and Planetary Sciences, Massachusetts Institute of Technology, Cambridge, MA, USA. [8]Physical and Life Sciences Directorate, Lawrence Livermore National Laboratory, Livermore, CA, USA. [9]Sorbonne Université, CNRS, UMR 7093, Laboratoire d'Océanographie de Villefranche-sur-Mer (LOV), Villefranche-sur-Mer, France. [10]Department of Earth, Ocean and Ecological Sciences, University of Liverpool, Liverpool, UK. [11]Department of Biological Sciences, University of Southern California, Los Angeles, CA, USA. [12]Biology Department, San Diego Mesa College, San Diego, CA, USA. [13]Faculty of Biology, Technion – Israel Institute of Technology, Haifa, Israel. [14]Department of Biological Sciences, California State University, Long Beach, CA, USA. [15]Department of Marine Chemistry and Geochemistry, Woods Hole Oceanographic Institution, Woods Hole, MA, USA. [16]Department of Oceanography, Texas A&M University, College Station, TX, USA. [17]School of Natural and Environmental Sciences, Newcastle University, Newcastle upon Tyne, UK. [18]Santa Fe Institute, Santa Fe, NM, USA. [19]Adobe, San Jose, CA, USA. [20]GEOMAR Helmholtz Centre for Ocean Research, Kiel, Germany. [21]Laboratory of Applied Evolutionary Biology, Department of Medical Microbiology, Academic Medical Centre, University of Amsterdam, Amsterdam, The Netherlands. [22]School of Oceanography, University of Washington, Seattle, WA, USA. [23]Institut de Biologie, École Normale Supérieure, Paris, France. [24]These authors contributed equally: Stephen J. Beckett, David Demory. ✉e-mail: beckett@umd.edu; david.demory@obs-banyuls.fr; jsweitz@umd.edu

