## [Peer Review File · Nature Communications]

Disentangling top-down drivers of mortality underlying diel population dynamics of *Prochlorococcus* in the North Pacific Subtropical GyreREVIEWER COMMENTS

Reviewer #1 (Remarks to the Author):

In this study, Beckett and co-authors developed a multitrophic community ecology model (ECLIP) to analyze the impacts of top-down and bottom-up controls in *Prochlorococcus* dynamics at a daily scale. Specifically, authors interrogated the contribution of both viral lysis and grazing to the overall mortality of *Prochlorococcus*. Interestingly, the authors found that grazing, rather than viral induced lysis, dominates *Prochlorococcus* mortality. Furthermore, *Prochlorococcus* losses were not completely explained by these mechanisms, but the model-inference suggests that there are other sources of *Prochlorococcus* losses.

This study contributes to a better understanding of the fate of primary production in oligotrophic surface ocean and consolidates the results of Mruwat et al., 2021.

I don't have any major comments. I consider that the manuscript is overall well-written, the model is robust, and the discussion comprehensively integrates the new findings with the inherent limitations of the model. I would like to see in the future how other measurements from different oligotrophic regions fit into the model described in this manuscript.

A minor comment related to the previous idea is whether there is already similar data generated from other regions that can be incorporated into the model to increase its robustness. The authors mentioned in the discussion (Lines 400-402) that: "Low levels of viral-induced *Prochlorococcus* (and *Synechococcus*) mortality have also been found in the Sargasso Sea (Matteson et al., 2013), and the Mediterranean and Red Sea (Mruwat et al., 2021)." I don't recall reading about these measurements in the Mediterranean and Red Sea from the cited paper (Mruwat et al., 2021). Sorry if I am missing something, but if similar measurements have already been taken, it might be worth incorporating them in this manuscript or in a follow up study.

Lastly, please consider using a similar format to fit the references within the text. I suggest keeping them at the end of the sentences unless it is necessary to embed them due to multiple examples or results being cited in the same sentence.

Reviewer #2 (Remarks to the Author):

Beckett et al., describe a mathematical model of Prochlorococcus, virus, grazer dynamics that seeks to replicate the dynamics of these organisms observed in a single time-series cruise in the North Pacific gyre. The paper is well written, modelling clearly described and has clear and well-presented figures and data. However, I am unsure of the logic of conducting this work or understanding what we have learnt from this model that we didn't already know from observations alone. Specifically I do not follow the logic of 'training' a model on data to then show you can replicate that data with the model. I would then expect to learn something new from the model by applying it to a situation where we do not have data! Or by testing the limits of the model to other interesting phenomena as I describe below. Thus, I do not believe the analyses presented are significant enough to justify publication in this journal.

The two main findings of the paper are 1) Grazing represents the predominant top-down control on Prochlorococcus and 2) Incorporation of light dependent viral traits allow a more representative modelling of virus dynamics. Unfortunately, both of these have already been reported in the literature by 1) Mruwat et al., 2021 and 2) Demory et al 2020 and Liu et al 2019. Thus, the main findings of this paper are not novel. If the novelty lies in the model formulation, the authors should make more effort to test other model formulations to explain why this one is superior.

In addition, I do not understand how this model can be considered "mechanistic" (line 137). How is the model mechanistic if parameter values are fitted to data. Nor do I believe it replicates the dynamics of viral and grazing populations very well. Figure 2, the model outputs do not fit the data well at all. Figure 6 is has a much better fit to %infected cells but not to free virus or grazer abundance. The authors state (line 260) "In sum, a range of nonlinear mathematical models including feedback between cyanobacteria, cyanophage and grazers can jointly recapitulate multi-trophic population dynamics in the NPSG". I cannot see justification for this claim. Where do the authors show this? Can they quantify this?

Lastly, I can think of several alternative avenues to which to apply the model which would improve the paper. 1) Can the model be applied to data from Hunter-Cevera et al., 2014, Hunter-Cevera et al 2016 or Fowler et al., 2020 10.1073/pnas.1918439117? This would expand the usefulness to data the model had not been trained on. 2) Under which

conditions can viral pressure exceed grazing? 3) What about also using the model to study phenomena that cannot be measured/observed. E.g. trade-offs in infection and grazing. Are infected cells equally grazed as non-infected cells? How does viral infection interact with other forms of mortality. E.g. many bacterial defense systems involve cell suicide. Can the model enable us to understand the significance of this process in host dynamics? Can it explain differences in encounter rates and infection rates?

In addition, I have below specific comments for the authors to address:

- 1) Line 147 -152. The description of the model here swaps between formulation and actual observations. Can the authors clarify which is which.
- 2) Line 167: Can the authors give a much clearer description of how the Prochlorococcus cell division rate works. Specifically how it differs from that used by Hunter-Cevera above. I notice the authors have access to cell size data, but I am unclear how this is used in the model.
- 3) Line 169: Pro loss rate is dependent on total Pro abundance. How is it dependent?
- 4) Line 173: What is the parameter n in the calculation average age duration? Is this defined somewhere?
- 5) Figure 3: The plots are not labelled a-j – though it is obvious from the legend.
- 6) Line 329-332: Despite reading several times, I cannot understand what this means? Can the authors clarify for a broader audience?

Response to reviewers: Disentangling top-down drivers of mortality underlying diel population dynamics of *Prochlorococcus* in the North Pacific Subtropical Gyre.
NCOMMS-23-20204

REVIEWER COMMENTS

Reviewer #1 (Remarks to the Author):

In this study, Beckett and co-authors developed a multitrophic community ecology model (ECLIP) to analyze the impacts of top-down and bottom-up controls in *Prochlorococcus* dynamics at a daily scale. Specifically, authors interrogated the contribution of both viral lysis and grazing to the overall mortality of *Prochlorococcus*. Interestingly, the authors found that grazing, rather than viral induced lysis, dominates *Prochlorococcus* mortality. Furthermore, *Prochlorococcus* losses were not completely explained by these mechanisms, but the model-inference suggests that there are other sources of *Prochlorococcus* losses.

This study contributes to a better understanding of the fate of primary production in oligotrophic surface ocean and consolidates the results of Mruwat et al., 2021.

I don't have any major comments. I consider that the manuscript is overall well-written, the model is robust, and the discussion comprehensively integrates the new findings with the inherent limitations of the model. I would like to see in the future how other measurements from different oligotrophic regions fit into the model described in this manuscript.

We thank Reviewer 1 for their encouraging comments and positive feedback. We are also optimistic that the current model framework will be applicable to other regimes - particularly as field-based campaigns increasingly use concurrent methods to assess grazer and viral impacts together. We have added a sentence to this effect in the Discussion, as follows:

“ Field-based campaigns that concurrently measure grazing and viral impacts will help further constrain the balance between viral- and grazing-induced mortality.”

A minor comment related to the previous idea is whether there is already similar data generated from other regions that can be incorporated into the model to increase its robustness. The authors mentioned in the discussion (Lines 400-402) that: “Low levels of viral-induced *Prochlorococcus* (and *Synechococcus*) mortality have also been found in the Sargasso Sea (Matteson et al., 2013), and the Mediterranean and Red Sea (Mruwat et al., 2021).” I don't recall reading about these measurements in the Mediterranean and Red Sea from the cited paper (Mruwat et al., 2021). Sorry if I am missing something, but if similar measurements have already been taken, it might be worth incorporating them in this manuscript or in a follow up study.

We thank the reviewer for this feedback and welcome the opportunity to clarify similarities and differences between this and other data.

First, we are not aware of another field campaign that has concurrent, high-temporal resolution time series measurements of host, infected host, virus, and grazer abundances. Hence, although we can compare infection rates between regions, we are unable to directly

compare relative mortality rates using time-series methods. As context, Mruwat et al. 2021 reports iPolony measurements from the Mediterranean (Supp Fig1) and Red Sea (Supp Fig 4). There is also related, Red Sea data published in Fig 5 of Maidanik et al. (ISME J, 2022). This paper is now cited in our revision. Additional iPolony measurements for the North Pacific are reported in Carlson et al. (Nat. Micro. 2022) showing a Northern transition zone with increased viral infection rates associated with gradients of environmental conditions. The Discussion includes a description of the typical ranges of cyanophage infection using the iPolony based method. However, previous work on inferring per-capita viral lysis from iPolony data used a simplified model that estimates daily mortality based on assumed latent periods and division rates, in the absence of time series data and in the absence of grazer effects. The takeaway here is that our time-series based inference of levels of viral mortality is consistent with work in the Mediterranean and Red Sea but is significantly lower than that found in the Northern transition zone. This brings us to a relevant point and new analysis.

While we cannot evaluate the methodology on additional field timeseries (none exist that we are aware of with relevant joint measurements), we have conducted a new sensitivity analysis to assess the robustness of our findings with respect to inferred ecological life-history traits. Our intention is to explore if the finding of relatively low viral mortality was inevitable in the model or is a function of life history traits which could vary across oceanic regions and context. We find that our model findings are robust to small changes in parameters. However, as noted above, by individually varying life-history traits to a greater extent we find that viral induced mortality could rival, or potentially exceed, that of grazer induced mortality. Mortality outcomes are most sensitive to variation in contact rates (grazer clearance rates and viral adsorption rates). For example, decreasing grazer contact rates by a factor of 4x or increasing viral contact rates by 4x shifts model dynamics to a regime in which viral-induced cell lysis is on par with that of grazing induced mortality. This point is emphasized in the Results and Discussion.

Lastly, please consider using a similar format to fit the references within the text. I suggest keeping them at the end of the sentences unless it is necessary to embed them due to multiple examples or results being cited in the same sentence.

We thank the reviewer and have gone back through our manuscript, keeping references at the end of sentences whenever possible and reformatting according to Nature Communications preferred style.

Reviewer #2 (Remarks to the Author):

Beckett et al., describe a mathematical model of Prochlorococcus, virus, grazer dynamics that seeks to replicate the dynamics of these organisms observed in a single time-series cruise in the North Pacific gyre. The paper is well written, modelling clearly described and has clear and well-presented figures and data. However, I am unsure of the logic of conducting this work or understanding what we have learnt from this model that we didn't already know from observations alone. Specifically I do not follow the logic of 'training' a model on data to then show you can replicate that data with the model. I would then expect to learn something new from the model by applying it to a situation where we do not have data! Or by testing the limits of the model to other interesting phenomena as I describe

below. Thus, I do not believe the analyses presented are significant enough to justify publication in this journal.

We thank the reviewer for their feedback and welcome the opportunity to clarify our intent and methodology. Following this, we address the reviewer's concern regarding the novelty of our findings.

Intent: In this study we developed mathematical population models of ecological interactions to infer mechanistic feedback rules and quantitative life history traits sufficient to explain the joint, observed dynamics of *Prochlorococcus*, infected *Prochlorococcus*, specific virus types known to infect *Prochlorococcus*, and specific abundances of grazers expected to consume *Prochlorococcus*. This is the only aggregation of time-series combining these measurements at a focal site. Model-data fitting is not guaranteed – it requires both a model that reasonably approximates the ecology of the system, and sufficient amounts of high quality data. It could be that our models are overly simplistic and/or we do not yet have sufficient constraints on life history traits to jointly and robustly fit mechanistic models to data with well-constrained priors on governing parameters. We emphasize that our approach focuses on inference using models developed based on our ecological understanding and mechanistic interactions between populations, rather than training a statistical 'black box' model to recognise and/or fit patterns in the datasets.

Methodology: We used a Bayesian MCMC approach that combines prior constraints on parameters as a means to explore the state space of mechanistic parameters that - when included as part of nonlinear mathematical models - generate timeseries that can be directly compared to observations. A full accounting of our Bayesian MCMC analysis using the Turing package in Julia is included in the SI - showing convergence, robust estimation of priors, and negative log likelihood ranges across model fits. We also showed in the revision that we could not recapitulate diel variation in infected cell abundances when only cellular traits varied with light, but only when also including diel variation in adsorption rate (see our Fig 6). The difference in model fits indicates that diel adsorption is an ecologically relevant process required to explain diel cycles of cyanophage infection at population scales. This type of methodology further reinforces our intent to develop a mechanistic model that captures both quantitative and qualitative features of multiple time-series, and then use inferred parameters and features of the models that are not directly observed in the data to advance ecological understanding.

The two main findings of the paper are 1) Grazing represents the predominant top-down control on *Prochlorococcus* and 2) Incorporation of light dependent viral traits allow a more representative modelling of virus dynamics. Unfortunately, both of these have already been reported in the literature by 1) Mruwat et al., 2021 and 2) Demory et al 2020 and Liu et al 2019. Thus, the main findings of this paper are not novel. If the novelty lies in the model formulation, the authors should make more effort to test other model formulations to explain why this one is superior.

We welcome the opportunity to clarify the novel, ecologically relevant conclusions of the present paper, clarify the extent to which prior work could constrain the quantitative levels of top-down loss of *Prochlorococcus* in the NPSG (or elsewhere), and emphasize the novel methodological elements of the study.

Grazing inference in context: The papers referred to by the Reviewer do not include grazing dynamics. Here, we have a model that jointly includes grazers AND virus dynamics. We then fit the model to *in situ* data using a state-of-the-art MCMC method, and directly estimate the levels of grazing and viral-induced mortality so as to compare these two levels. If we had not done so, then we would have ascribed whatever was 'missing' to the other component - here we addressed the harder task of jointly estimating both features simultaneously. We are unaware of any model that does so in field settings (a key novel element) as applied to joint time series measurements (again, no other field campaign has such joint data) to make robust claims about relative mortality (in contrast to other approaches that make assumptions about one or the other mortality source). In doing so, we also point out that the prior work at this field site used a quota approach which found that grazers could potentially account for all *Prochlorococcus* daily mortality but could also represent a relatively insignificant fraction of daily mortality - this unconstrained mortality estimate leaves the question of what drives the daily balance of growth and loss unresolved. Here, the central biological conclusion is our finding that grazers account for >90% of mortality, viruses account for <5% of mortality and that other, density-dependent factors other than top-down loss account for ~5% of mortality.

Viral lysis, in context: The present studies significantly advances our understanding of the limits of viral lysis to *Prochlorococcus* mortality. First, the papers we already cite (and in some cases are coauthors on) measured viral infection levels using the iPolony method or inferred viral mortality from time series in the absence of grazers. In contrast, here we jointly estimate mortality sources to reach our conclusion that grazing and not viral lysis predominates in this location. Second, our incorporation of light-dependent adsorption was found previously in lab-based experiments - but that doesn't mean such light-dependent adsorption is necessarily critical to recapitulate observed dynamics in the NPSG. The fact that light-dependent adsorption is critical to recapitulate diel oscillations in infected cells suggests that viral trait plasticity is ecologically relevant and not just an incidental feature of viral adsorption that can be observed in lab settings.

Methodology: As described in more detail below, the present analysis goes beyond the methods of Mruwat et al. 2021 to jointly consider viral- and grazing- mortality on *Prochlorococcus* in the NPSG; and to consider how infected cell measurements (such as via the new iPolony method) might be incorporated into data-driven ecological models. Mruwat et al., 2021 do not explicitly analyze grazing data, but hypothesize it is the likely driver of *Prochlorococcus* mortality given estimated low viral-induced lysis through a steady-state type analysis. Moreover, unlike the model fits of Demory et al. and Lui et al. our findings relate to *in situ* field data in the North Pacific rather than controlled laboratory experiments that do not include grazers. We are unaware of any other aggregated field campaign for which we could apply this type of approach. The joint MCMC model-data inference represents the first time (that we are aware) of a multi-trophic model being used to quantitatively fit high-temporal resolution data of cells, infected cells, viruses and grazers. This methodology along with open-source code represents an advance that can be used as part of future field campaigns that collect concurrent measurements of top-down effects. The entire package including source code, data, analysis, and figures is available via an open-access Zenodo archive here: <https://doi.org/10.5281/zenodo.10127185>.

In addition, I do not understand how this model can be considered “mechanistic” (line 137). How is the model mechanistic if parameter values are fitted to data.

Our use of the term ‘mechanistic’ refers to the fact that we have built a mathematical representation of interactions that include cell division, viral adsorption, viral lysis, grazer uptake and division, as well as decay of each population. These mechanisms include parameters with ecological interpretations. We are not fitting curves to data (whether linear, sinusoidal, quadratic, or otherwise). We have added a sentence when we first use the term mechanistic to clarify this point:

“We developed a mechanistic mathematical model of an Ecological Community driven by Light including Infection of Phytoplankton (ECLIP). The mechanistic model is driven by parameterized interactions with ecological interpretations. Our model includes dynamics of *Prochlorococcus*, grazers, and viruses, as well as *Prochlorococcus* division and loss, where the loss arises due to a combination of grazing, viral lysis, and other factors.”

Nor do I believe it replicates the dynamics of viral and grazing populations very well. Figure 2, the model outputs do not fit the data well at all. Figure 6 is has a much better fit to %infected cells but not to free virus or grazer abundance. The authors state (line 260) “In sum, a range of nonlinear mathematical models including feedback between cyanobacteria, cyanophage and grazers can jointly recapitulate multi-trophic population dynamics in the NPSG”. I cannot see justification for this claim. Where do the authors show this? Can they quantify this?

We note that if the model fit every data point perfectly, it would be evidence of a severe example of over-fitting. Instead, the point of Figure 2 is to show that regardless of the level of grazer generalism, we are able to infer dynamic ecological models with similar dynamics. Hence, our findings are robust to variation in grazer generalism. We are glad that the Reviewer agrees that Figure 6 has improved fits to infected cells - though given measurement noise and biological variation we do not and should not expect perfect fits. Note that MCMC performance is assessed in Supp Figure S5 and fit quality is assessed in Supp Figure S9. Further, as explained in Supp Text S1 and Table S1, neither the virus nor the grazer abundances had statistical evidence of periodicity using the ‘RAIN’ non-parameteric inference package. Thus, the variation in the timeseries of either grazer and virus abundances should not be interpreted as indications of the model missing periodicity.

Lastly, I can think of several alternative avenues to which to apply the model which would improve the paper. 1) Can the model be applied to data from Hunter-Cevera et al., 2014, Hunter-Cevera et al 2016 or Fowler et al., 2020 10.1073/pnas.1918439117? This would expand the usefulness to data the model had not been trained on.

The aim of the study is to infer ecological rates via model-data integration that jointly considers impacts of grazing and viral lysis. Hence, our model framework is an alternative and complementary approach to the size-structured models developed in e.g. Hunter-Cevera et al., 2014, Hunter-Cevera et al 2016, Ribalet et al. 2015. These approaches are data-driven, i.e., they implicitly incorporate environmental-physiological feedback to infer division and mortality rates from differences in timeseries of cell-size and cell abundance –

however, they do not distinguish the source of mortality (i.e., they are measuring what is left over after accounting for measured growth and ascribing that to total mortality, irrespective of its mechanistic source). The papers referred to by the Reviewer do not contain the joint measurements of cell abundances, viral abundances, and grazer abundances needed to make the joint inferences of mortality (and to compare relative mortality). We hope that the present work will inspire other groups to assess and compare viral and grazing induced mortality in other field and experimental settings.

2) Under which conditions can viral pressure exceed grazing?

We are not sure if the authors are referring to claims in the literature or as estimated using our model. As context, we note that viruses have been suggested to constitute up to 40% of phytoplankton mortality e.g., (Brussaard et al. 2005; Payet and Suttle, 2013 amongst many) rivaling if not exceeding that of grazing. Yet, such claims are based on making strong assumptions regarding viral particle decay. In this revision, we added a new Results section in which we assess the robustness of model findings to variations in parameters that we inferred in our Bayesian MCMC method. In particular we identify how variations in life-history attributes of the viruses and/or grazers can influence viral and/or grazing pressure. Thus, viral pressure can equal if not exceed that of grazing with modest changes in clearance and/or viral adsorption rate. Hence, we have re-emphasized that although we find robust evidence that grazing losses predominate over viral induced lysis in the NPSG time series studied here, that such a relative ordering need not apply to other regions or other virus-host systems. The updated Results section and figures provide more details, including references to recent observations in the North Pacific transition zone at single location/time estimates where viral infection rates can be far higher than observed in the NPSG (Carlson et al. 2022).

3) What about also using the model to study phenomena that cannot be measured/observed. E.g. trade-offs in infection and grazing. Are infected cells equally grazed as non-infected cells? How does viral infection interact with other forms of mortality. E.g. many bacterial defense systems involve cell suicide. Can the model enable us to understand the significance of this process in host dynamics? Can it explain differences in encounter rates and infection rates?

We agree that these are important questions on the frontiers of microbial ecology – and that the model developed here could be used to address these questions in the future. Such modifications are beyond the scope of the current work and likely require multiple applications of the present model framework to make cross-system comparisons. We share the reviewers' enthusiasm for future exploration of such questions and have discussed multiple directions for future work in the Discussion.

In addition, I have below specific comments for the authors to address:

1) Line 147 -152. The description of the model here swaps between formulation and actual observations. Can the authors clarify which is which.

In Lines 147-154, we discuss two separate ideas.

1) The 2015 NPSG system has heterotrophic nanoflagellates, phototrophic nanoflagellates, and mixotrophic nanoflagellates. However, we have two types of measurements: (a)

heterotrophic nanoflagellates and (b) a combined measurement of phototrophic and/or mixotrophic nanoflagellates. As we are unable to distinguish between mixotrophic and phototrophic nanoflagellates and because mixotrophic grazing pressure is presumed to be less intense than grazing pressure from heterotrophic nanoflagellates we have made a choice to only include heterotrophic nanoflagellate grazing in our model and data-integration efforts.

2) We do not have abundance measurements of all the types of prey items that heterotrophic nanoflagellates consume, yet we may expect them to consume other organisms (within their size-range) present in the system. We have incorporated this uncertainty into our modeling framework via the generalism parameter γ , which effectively defines the relative amount that *Prochlorococcus* consumption constitutes within the heterotrophic nanoflagellates diet. We have quantified this by defining what we call the degree of grazer generalism, shown in Equation S20.

Original text (Line 147-152):

“To assess this uncertainty we investigated six grazer strategies, ranging from a “specialized” grazer class exclusively consuming *Prochlorococcus* cells ($\gamma = 0 \text{ day}^{-1}$) to models with increasing levels of generalism ($\gamma = 0.01$ to $\gamma = 0.5 \text{ day}^{-1}$) representative of grazers consuming additional prey, e.g., heterotrophic bacteria which are not explicitly integrated into the model. Mixotrophic nanoflagellates (Caron, 2017) were observed, but contribute less to the grazing pressure on the bacterial community compared to heterotrophic nanoflagellates (Connell et al., 2020). As it was not possible to differentiate abundance measurements of mixotrophic nanoflagellates from phototrophic nanoflagellates (Connell et al., 2020), we focus on grazing by heterotrophic nanoflagellates.”

To make this clearer for readers we have edited as follows:

“To assess this uncertainty we investigated six grazer strategies, ranging from a “specialized” grazer class exclusively consuming *Prochlorococcus* cells ($\gamma = 0 \text{ day}^{-1}$) to models with increasing levels of generalism ($\gamma = 0.01$ to $\gamma = 0.5 \text{ day}^{-1}$) representative of grazers consuming additional prey, e.g., heterotrophic bacteria which are not explicitly integrated into the model. Mixotrophic nanoflagellates (Caron, 2017) were observed, but contribute less to the grazing pressure on the bacterial community compared to heterotrophic nanoflagellates (Connell et al., 2020). As it was not possible to differentiate abundance measurements of mixotrophic nanoflagellates from phototrophic nanoflagellates (Connell et al., 2020), we focus only on grazing by heterotrophic nanoflagellates.”

2) Line 167: Can the authors give a much clearer description of how the *Prochlorococcus* cell division rate works. Specifically how it differs from that used by Hunter-Cevera above. I notice the authors have access to cell size data, but I am unclear how this is used in the model.

In this formulation of the model, cell size data is not incorporated explicitly. We implicitly use cell-size data through a modeled division rate that is incorporated into our model. In particular, Ribalet et al. 2023 use cell-size data to characterize time-dependent *Prochlorococcus* division rates. In our study, we fit a sine-wave function to the division rates estimated by Ribalet et al. 2023 as an effective approximation of the division rate. Future work could extend the ECLIP framework to include an explicit representation of size structure in addition to the concurrent forces of viral lysis and grazing.

3) Line 169: Pro loss rate is dependent on total Pro abundance. How is it dependent?

We have incorporated a density-dependent loss term in our model in order to introduce a carrying-capacity like feature to our model to represent niche competition. In essence, the higher the abundance of Pro., the higher the amount of mortality which scales with total Pro abundance i.e., as shown in Equation 1 we model Pro. per cell loss rate dependent as: $m_P \cdot (S+I)$. Such higher order loss terms are standard in ecosystem modeling.

4) Line 173: What is the parameter n in the calculation average age duration? Is this defined somewhere?

We believe this may be a formatting misinterpretation via clipping of the word average between lines 172-173 of the reviewed manuscript. In line 173 η corresponds to the average rate of lysis by infected cells, such that (as we define) the inverse of η is the average duration of infection – corresponding to the latent period. To avoid confusion we have prevented the clipping of the word average.

5) Figure 3: The plots are not labelled a-j – though it is obvious from the legend.

We thank Reviewer 2 for spotting this and have added labels to Figure 3 in our revision.

6) Line 329-332: Despite reading several times, I cannot understand what this means? Can the authors clarify for a broader audience?

We have rewritten the content on lines 329-332 for clarity.

The original text reads:

“For viral lysis, one-factor estimates using encounter theory do not constrain daily loss rates, given that contact-limited lysis rates are compatible with nearly 100% of observed *Prochlorococcus* loss; but decreases in contact rates, efficiency of adsorption, and inefficiency of infection post-lysis lead to poorly constrained lower limits.”

The new updated text reads:

“For viral lysis, one-factor estimates using encounter theory do not constrain daily loss rates. For example, if lysis was limited by contact with host cells, then observed viral abundances could account for nearly 100% of observed *Prochlorococcus* loss. Conversely, estimated daily loss rates could decrease to nearly 0% if contact rates were significantly lower than biophysical limits suggest, adsorption was inefficient, or adsorption did not necessarily imply a successful infection because some phage were non-infective and/or defective.”

REVIEWERS' COMMENTS

Reviewer #1 (Remarks to the Author):

The authors have satisfactorily addressed my comments. I consider their work to be an important contribution to advancing more accurate ecological models of picocyanobacterial populations by integrating longitudinal data on *Prochlorococcus* cell abundance, infected *Prochlorococcus*, grazers, and viruses. This contribution is likely to stimulate further research in other photosynthetic models and regions. Additionally, this study coherently suggests the existence of overlooked potential mortality processes, posing an intriguing question for further investigation.

Reviewer #2 was satisfied and had no more comments.

Reviewer #3 (Remarks to the Author):

I have been asked to review the modelling component of the paper, recognizing that I have expertise in modelling virus and other population dynamics, but not of cyanobacteria.

Overall the modelling is well conducted and the model well described.

1. I was surprised to see the inclusion of quadratic loss terms for viral particles: I can see that this is well established in this community. Interestingly it is not normally used by modellers studying dynamics of viral dynamics in human/animal diseases (that includes very well known researchers); in our own work (on phage mediated transfer of antimicrobial resistance genes) we resolved the well-known problem of exclusion/extinction when virus loss terms are linear by including simple terms for rapid virus/prey co-evolution, a mechanism that also stabilizes these dynamics, and is well known from the work of Delbrück and Luria. It would appear that greater communication between different communities of virus dynamics modellers would be implicated!

In terms of this paper, I would prefer to see the word 'empirical' on L193 i.e.

"Grazer and viral losses are characterized by (empirical) nonlinear loss terms"

as this is an empirical rather than mechanistic approach - but this is not essential and leave it at the discretion of the authors.

Similarly, the authors should add to discussion a sentence or two that viral/host co-evolution is likely to be extremely rapid (given viral mutation rates) and impact upon population dynamics (possibly profoundly, as we have found), but of course such analysis is out of scope for this article and could be the basis of future work (hence 1-2 sentences of Discussion).

***Editor note:** the reviewer suggested we share the following reference in support of this comment, in case it is helpful - Arya S, Todman H, Baker M, Hooton S, Millard A, Kreft JU, Hobman JL and Stekel DJ. 2020. A generalised model for generalised transduction: the importance of co-evolution and stochasticity in phage mediated antimicrobial resistance transfer. FEMS Microbiology Ecology: fiae100.

2. I have some technical concerns about Figure S2 that need to be addressed. The authors state that they have used wide uniform priors (which is concordant with Table S3). Two changes are needed: (i) The parameter names (μ_{ave} , δ_{μ} and δ_t) need to be added either to the axis labels or to the Figure caption. (ii) More importantly, the orange lines corresponding to the priors are not correct: they are wiggly, especially the middle panel! These priors should be FLAT in their range, and it might work better if the y-axes were probability density. These must be redrawn!

3. It was really nice to see the joint posterior estimates of correlated parameters in Figure S7. Have the authors checked that the 'best' parameter estimates from the joint posteriors may be different from the 'best' estimates from the marginal posteriors? (I appreciate I am using the word 'best' ver loosely here as it strictly isn't a Bayesian concept). I say this because this can happen with banana-shaped joint posteriors (again something we have

seen in our own work) and where the use of marginal posteriors can be misleading.

***Editor note:** the reviewer suggested we share the following reference in support of this comment, in case it is helpful - Herman, D., Thomas, C.M. and Stekel, D.J. 2011. Global transcription regulation of RK2 plasmids: a case study in the combined use of dynamical mathematical models and statistical inference for integration of experimental data and hypothesis exploration. *BMC Systems Biology* 2011, 5:119.

Response to reviewers: Disentangling top-down drivers of mortality underlying diel population dynamics of *Prochlorococcus* in the North Pacific Subtropical Gyre. NCOMMS-23-20204A

Response to Reviewer 3:

1. I was surprised to see the inclusion of quadratic loss terms for viral particles: I can see that this is well established in this community. Interestingly it is not normally used by modellers studying dynamics of viral dynamics in human/animal diseases (that includes very well known researchers); in our own work (on phage mediated transfer of antimicrobial resistance genes) we resolved the well-known problem of exclusion/extinction when virus loss terms are linear by including simple terms for rapid virus/prey co-evolution, a mechanism that also stabilizes these dynamics, and is well known from the work of Delbrück and Luria. It would appear that greater communication between different communities of virus dynamics modellers would be implicated!

>> We thank the reviewer for this point. We agree that rapid coevolution can act to stabilize population dynamics, but that including rapid coevolution and their corresponding phage-bacterial infection networks is beyond the scope of this study. In our approach, we seek to avoid introducing structural bias between the grazer and viral predators, while allowing for model closure. Our approach also has the benefit of not introducing additional parameters, which might make the parameter space more difficult to infer via MCMC methods. We agree with the reviewers' vision of promoting ongoing and future efforts to communicate viral dynamics, and how to model them, across scales – and have included an additional sentence and references in the Discussion section to address this point (described below).

In terms of this paper, I would prefer to see the word 'empirical' on L193 i.e.

"Grazer and viral losses are characterized by (empirical) nonlinear loss terms"

as this is an empirical rather than mechanistic approach - but this is not essential and leave it at the discretion of the authors.

>> We thank the reviewer for their feedback, however after some discussion we have chosen to leave the text here as is. We have revisited the issue of coevolutionary dynamics in the Discussion to address the issues raised by the reviewer.

Similarly, the authors should add to discussion a sentence or two that viral/host co-evolution is likely to be extremely rapid (given viral mutation rates) and impact upon population dynamics (possibly profoundly, as we have found), but of course such analysis is out of scope for this article and could be the basis of future work (hence 1-2 sentences of Discussion).

*Editor note: the reviewer suggested we share the following reference in support of this comment, in case it is helpful - Arya S, Todman H, Baker M, Hooton S, Millard A, Kreft JU, Hobman JL and Stekel DJ. 2020. A generalised model for generalised transduction: the importance of co-evolution and stochasticity in phage mediated antimicrobial resistance transfer. FEMS Microbiology Ecology: fiae100.

>> We agree with the reviewer that coevolution at the microbial scale occurs on rapid timescales and could have a role in determining ecological stability, and also that such a model

is beyond the scope of this work. Indeed, we note that members of our team have previously worked on multiple such models e.g., Weitz et al., 2005, PNAS, doi: [10.1073/pnas.0504062102](https://doi.org/10.1073/pnas.0504062102) ; Beckett et al., 2013, RSIF, doi: [10.1098/rsfs.2013.0033](https://doi.org/10.1098/rsfs.2013.0033) ; Rodriguez-Gonzalez et al. 2020, mSystems, doi: [10.1128/msystems.00756-19](https://doi.org/10.1128/msystems.00756-19). We have added the following text (new text highlighted in yellow) to the Discussion section (new text starts Line 337, Page 10) to address this:

“However, ECLIP does not explicitly capture size-structured processes which are important drivers of growth (Hunter-Cevera et al., 2014; Hynes et al., 2015; Mattern et al., 2022; Ribalet et al., 2015) and other ecological interactions (Talmy et al., 2019b). Additionally, light-driven forcing of division does not fully account for variability in processes such as nutrient content (Lopez et al., 2016; Muratore et al., 2022; Vislova et al., 2019), and metabolic state (Muratore et al., 2022). While these attributes are not specifically modeled, they may have bearing on inferring life-history traits. Further, we note that (co)evolutionary dynamics within microbial systems, especially with respect to the viruses, can occur on rapid timescales (Marston et al., 2012), which has the potential for strain-level differentiation in life-history traits and infection networks which can alter and potentially control population dynamics (Weitz et al., 2005; Beckett et al., 2013; Choua et al. 2020; Arya et al., 2020). ECLIP provides a complementary framework for understanding marine microbial ecology; and we hope future efforts will attempt to blend these types of models.”

- Marston et al. 2012. [10.1073/pnas.1120310109](https://doi.org/10.1073/pnas.1120310109)
- Choua et al., doi: [10.1016/j.jtbi.2020.110263](https://doi.org/10.1016/j.jtbi.2020.110263)
- Arya et al. 2020 (supplied by reviewer). [10.1093/femsec/fiaa100](https://doi.org/10.1093/femsec/fiaa100)

2. I have some technical concerns about Figure S2 that need to be addressed. The authors state that they have used wide uniform priors (which is concordant with Table S3). Two changes are needed: (i) The parameter names (μ_{ave} , δ_{μ} and δ_t) need to be added either to the axis labels or to the Figure caption. (ii) More importantly, the orange lines corresponding to the priors are not correct: they are wiggly, especially the middle panel! These priors should be FLAT in their range, and it might work better if the y-axes were probability density. These must be redrawn!

>> We thank the Reviewer for identifying this issue. We have amended Figure S2 inline with the suggestions from the reviewer. The figure now includes parameter symbols in labels, and the lines representing uniform priors have been corrected. We have amended the y-axis to represent Density to make this plot consistent with other posterior plots in the manuscript.

3. It was really nice to see the joint posterior estimates of correlated parameters in Figure S7. Have the authors checked that the 'best' parameter estimates from the joint posteriors may be different from

the 'best' estimates from the marginal posteriors? (I appreciate I am using the word 'best' ver[y] loosely here as it strictly isn't a Bayesian concept). I say this because this can happen with banana-shaped joint posteriors (again something we have seen in our own work) and where the use of marginal posteriors can be misleading.

*Editor note: the reviewer suggested we share the following reference in support of this comment, in case it is helpful - Herman, D., Thomas, C.M. and Stekel, D.J. 2011. Global transcription regulation of RK2 plasmids: a case study in the combined use of dynamical mathematical models and statistical inference for integration of experimental data and hypothesis exploration. BMC Systems Biology 2011, 5:119.

>> We thank the reviewer for raising this issue, and we agree that there are potential issues with parameter identifiability when using MCMC, which is why we looked more closely at correlated parameters (also see Fig S6). In our Bayesian inference approach, we simultaneously estimate the parameters, such that we take account of the whole distribution in evaluating model-data fits and in quantifying ecological process rates. Even if we have sloppy parameters the variability of those parameters is taken into account in our analysis. While we see correlation between some life-history parameters, we do not see evidence for multiple parameter sets occurring within each treatment. Our intent here was also to investigate if there were changes in parameter variation between different levels of grazer generalism, which we found to be small, with the exception of variation in grazer mortality (m_G). Hence, we have decided to retain our current treatment of posteriors and interpretation of fits.